# Offshoring Refugees: Colonial Echoes of the UK-Rwanda Migration and Economic Development Partnership

**Michael Collyer** [1,*] **and Uttara Shahani** [2]

1   Department of Geography, University of Sussex, Brighton BN1 9RH, UK
2   Refugee Studies Centre, Department of International Development, University of Oxford, Oxford OX1 3TB, UK; uttara.shahani@qeh.ox.ac.uk
*   Correspondence: m.collyer@sussex.ac.uk

**Abstract:** British proposals to forcibly deport asylum seekers to Rwanda have raised fierce opposition from across the political spectrum in the UK and internationally. These proposals differ from official practices of deportation as they have developed in liberal democracies since the 1970s. There are certainly some international parallels, such as Australia's 'Pacific Solution' of 'offshoring' asylum, which is often cited as an inspiration. Yet a much clearer precedent involving the forcible movement of people to countries where they have no personal or legal connection existed for many years in the British Empire. Colonial policies of forcible removal, relocation, displacement, and dispersal around the Empire are well established. We draw attention to these longer histories before investigating more recent cases of the dispersal of refugees within the British Empire in the twentieth century. In many cases, such forced dispersal concerned those who had been recognised as refugees who were interned and subsequently moved elsewhere in the Empire. Such policies were designed to prevent the arrival of refugees in the UK. These policies have provided inspiration for asylum practices in some postcolonial states—Israel is reported to have reached an agreement with Uganda and Rwanda to deport asylum seekers from Sudan and Eritrea, although these are not public. In this paper, we highlight how these colonial practices of forcible displacement of individuals inform the current agreement between the UK and Rwanda.

**Keywords:** relocation; deportation; dispersal; colonialism; deportation; asylum; British Empire; Rwanda; refugees

## 1. Introduction

In recent years, expulsion has become shorthand for analysis of the untrammelled exercise of political or economic power (Sassen 2014). More violent than exile, more permanent than imprisonment, the enforced removal of a person from somewhere where they want to be to somewhere it is assumed they do not has impacts beyond the often tragic individual stories involved. It is individual since the violence this process entails is directed at the body; it may cause physical harm and even death (Fekete 2011). Yet the work of expulsion goes far beyond the transfer of a certain number of bodies from one place to another. The broader significance of expulsion is that it reproduces and reinforces hierarchies between people and between places. In the context that we examine it here, expulsion refers to the enforced removal or deportation[1] of unwanted migrants and the colonial history of forced relocation of people in the British colonies comprising, amongst others, the enslaved, indigenous peoples, indentured labour, and refugees.

Deportation affects many more people than the limited numbers who can be deported. On an individual level, it increases fear and undermines the stability of the much larger number of people who may be considered deportable (De Genova 2002). In this article, we are concerned with the broader political effects of deportation. The process of reinforcing power dynamics that are historically embedded between people and places is almost

inevitably colonial. Deportation contributes to this process, as do certain other practices of migration control (Mayblin 2019) though our focus here is on deportation.

This article examines the 2002 UK-Rwanda Migration and Economic Development Partnership (MEDP), part of a flurry of primary and secondary legislation put before the UK Parliament in 2022 and 2023. In this case, the colonial antecedents of deportation practices go well beyond the broader effects of the reproduction of established power dynamics (though it also does that) to the repetition of exact practices and even forms of words from Britain's colonial past. We argue that the MEDP marks a break with the UK's post-1970s migration policy, which, with a brief exception from 2003–2010, involved the forced removal of relatively small numbers of people from the UK to their country of citizenship. In returning to practices of deportation and dispersal common, particularly towards the end of Britain's colonial period, the MEDP contributes to the extension of those colonial practices into the present day.

As this special issue attests, the analysis of the colonial basis for contemporary practices of asylum is now well established. The MEDP has already received very widespread critical attention, including commentary that has identified the colonial echoes of the policy (Akanga and Hughes 2022). Here, we contribute to this analysis in two ways, corresponding to the following two sections of this article. In the following section, we set out an analysis of the UK's post-1970s practice of deportation and enforced removals in order to highlight how the MEDP departs from this more recent practice.

The second section turns to the British colonial practices of forced relocation, of which the MEDP is a more obvious continuation. Moving people around the Empire and, concomitantly, forcing them to stay put was a cornerstone of British colonial rule that manifested in different forms of 'relocation' over the first and second British Empires. We briefly highlight some of the earlier histories drawing from the large body of secondary literature on the slave trade, transportation, indenture, and migration within the British Empire before turning our focus on the twentieth century (for which we have consulted both secondary sources and some primary sources) when practices of relocating refugees became common.[2] The aim is not to provide exhaustive coverage of British practices of relocation, of which there are many examples. Instead, our approach is to draw attention to a pattern of relocation, particularly the dispersal of refugees, which we argue strongly parallels contemporary plans to 'offshore' refugees. The dispersal of refugees to locations across the British Empire was relatively widespread during and immediately after the Second World War, and there is evidence that much more was considered or actively planned for. The conclusion draws these two sections together and places them once again in the colonial analysis of power hierarchies.

## 2. Enforced Removals from the UK and the UK-Rwanda Plan

The UK government's policy to forcibly remove selected rejected asylum seekers to Rwanda is set out in the UK-Rwanda Migration and Economic Development Partnership (MEDP), signed on 13th April 2022 (Gower et al. 2022). This agreement had been widely expected, and significant political opposition to it developed well before it was signed, largely due to the Nationality and Borders Act 2022, which provided the legislative basis for such agreements (in Schedule 4) and had been making its way through Parliament since July 2021. The Illegal Migration Bill 2022–23, which is going through Parliament at the time of writing, provides greater opportunities to implement the MEDP. Following the signature of the MEDP, opposition to forcible removal accelerated and included such establishment figures as King Charles III and the Archbishop of Canterbury, in addition to the coordinated efforts of human rights and refugee support organisations in the UK and internationally.

The first deportation under the MEDP was scheduled for 14th June 2022 and concerned 47 individuals. It was finally stopped as the plane was preparing to take off. Two separate legal challenges to the deportation plan were heard in September and October 2022. A final judgement on both cases, published at the end of December 2022, found that the Home



Secretary was legally entitled to order such deportations but allowed an appeal on a small number of points. At the time of writing, a final judgement on an appeal is expected in the second half of 2023. Although removals to Rwanda are suspended until that judgement is reached, individuals continue to receive notification that they are scheduled for removal to Rwanda. The Rwanda plan is a significant departure from contemporary deportation practice in the UK, which has developed since the 1970s. The central argument of this paper is that it marks a return to a pattern of forced mobility that was much more common during the British colonial period. This section briefly reviews current deportation practices before turning to a more detailed analysis of the Rwanda deportation plan and the legal challenges to it.

Enforced returns of rejected asylum seekers did not form a significant part of UK immigration control practice until 2001. In 2000, 2990 rejected asylum seekers were removed from the UK, a figure that had only grown very slowly since 1992 (when this statistic first appeared).[3] From 2000, enforced removals of rejected asylum seekers rose sharply, reaching a peak of 11,783 in 2004. This relatively static picture of removals during the 1990s occurred despite a significant rise in the number of asylum applications, the majority of which were rejected (Figure 1). In 2002, asylum applications to the UK peaked at 84,132. The political profile of asylum was sufficiently high that the response was directed from the Prime Minister's office rather than the Home Office. In September 2004, Tony Blair announced a target that monthly removals for rejected asylum seekers should exceed the number of rejections by the end of 2005 (BBC News 2004), ensuring the continued politicisation of statistics on enforced removals.

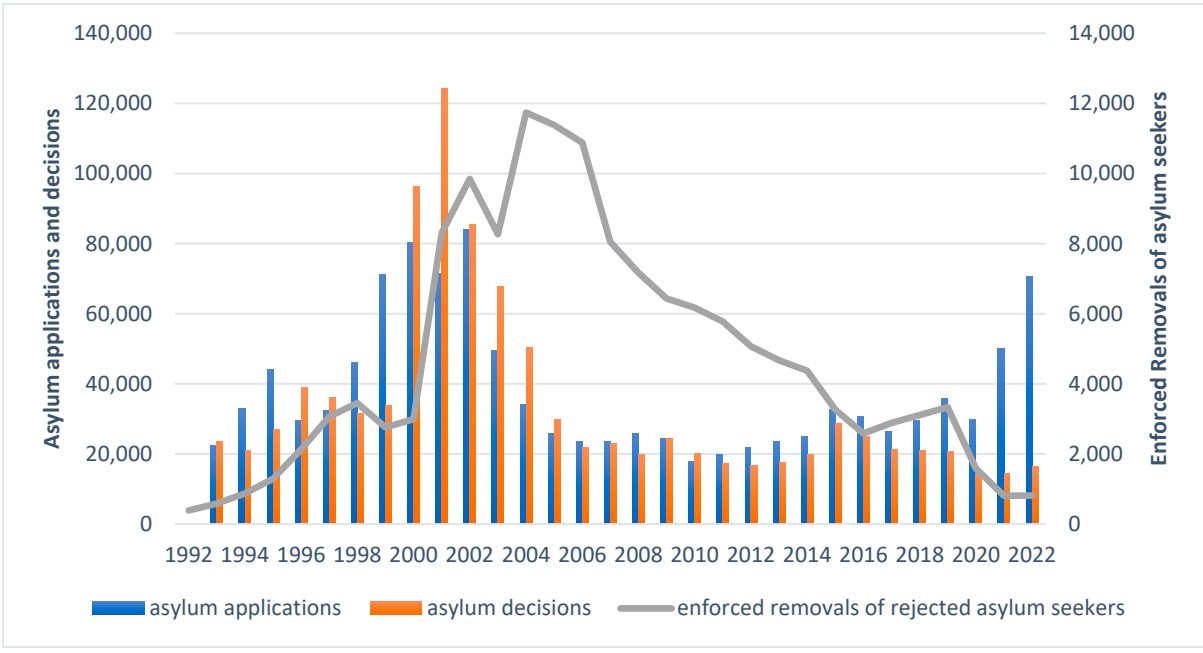

**Figure 1.** Asylum applications, decisions, and enforced removals of rejected asylum seekers from the UK 1992–2022 (source: produced by authors using data from Home Office (2013–2023).

In the space of a few years, between 2000 and 2004, the situation shifted from one in which forced returns were relatively low and only of theoretical importance to the UK's immigration enforcement to one in which they had grown substantially and become a dominant issue of national politics. This shift has been labelled the 'deportation turn' (Gibney 2008), a change exemplified by UK practice but one that was not unique to the UK. As Figure 1 shows, enforced returns of rejected asylum seekers were boosted in 2004 and 2005 by this political attention and have declined steadily since then.

The situation in 2022–23 has much in common with the deportation turn of the early 2000s, but the immediate response has been different. This helps explain the background

of the removals to Rwanda much more clearly, but it also helps to highlight the clear differences between the Rwanda plan and the responses twenty years earlier. Since 2010, applications for asylum began to grow, slowly at first and then more significantly in 2021. Over this period, decisions on asylum applications and removals of rejected asylum seekers continued to fall (Home Office 2023). This produced a situation in which, at the end of 2022, over 160,000 people had been waiting for an asylum decision for more than six months. Over this period, the UK has seen something of a 'deportation re-turn', with enforced removals dropping to levels last seen in the early 1990s, partly influenced by the Covid pandemic. This situation has been accompanied by the highly mediatised rise in the number of people arriving irregularly in the UK by small boats. Arrivals by small boat were virtually negligible in 2018, grew to 8466 by 2020, and became a dominant political issue in 2022 with 47,755 arrivals (Home Office 2023). This rise is partly explained by greater security at channel ports, which has shifted routes from comparatively safer and much less visible clandestine crossings in the backs of lorries (Home Office 2023). Still, the rapid rise in numbers and detailed media attention have helped fuel a narrative of crisis that has become directly associated with the current failings in the asylum system. For the first time in 20 years, asylum policy was once again a topic of sustained attention by the Prime Minister. The reduction in arrivals by small boats and corresponding attention to asylum was one of five 'pledges' set out by Rishi Sunak in January 2023 (BBC News 2023).

Removing people by force from one country and transferring them to another is a difficult process. In liberal democracies, there has historically been substantial public opposition to the forced removal of individuals, particularly those who have developed ties in local communities and neighbourhoods (Anderson et al. 2011). It is also extremely expensive, with a recent Home Office analysis putting the cost at £15,000 per individual removed in 2013 (Home Office 2013, p. 4)—at a time of significant constraints on public budgets, there was a clear financial reason for allowing forced deportations to fall, particularly at a time of very limited public concern. Despite this wide-ranging opposition to enforced removals, deportation performs a necessary role for states in particular and for the international state system in general (Hindess 2000). William Walters (2002, p. 265) has called this the 'international police' of populations. The enforced removal of an individual from a country where they are found to be illegally resident to their country of citizenship reinforces the bond between citizenship and territory, one of the foundations of the state system.

The large majority of deportations follow this pattern of forced removal from a country where an individual is deemed to be illegally resident to their country of citizenship. This presents certain geopolitical challenges since it requires the agreement of the state that is receiving the deported migrant. This may not be forthcoming, particularly where there is disagreement about the citizenship of the individual being deported or in cases where there are no diplomatic relations between the states concerned (Collyer 2012). Nevertheless, wealthy states have agreed on a barrage of readmission agreements, often under coercive terms, which are designed to smooth out this process. The only widespread exception to this pattern of removal directly to the individual's country of citizenship is covered by a range of 'safe third country' legislation. Since the Nationality, Immigration and Asylum Act 2002, (which was the legislation that facilitated the accelerating enforced removals from 2003 onwards), the government has been able to declare certain countries 'safe' for the purposes of claiming asylum. This facilitates accelerated removals to those countries. The label 'safe' can also be applied to countries that an individual may have come through before claiming asylum in the UK. In these very limited cases, an individual claiming asylum may be removed to a country where they are not a citizen if it can be demonstrated that they travelled through that country and had the opportunity to claim asylum there. They are, therefore, removed in the expectation that the receiving country will allow them to register an asylum claim. In the European Union, legislation first drawn up in 1990 under the Dublin Convention and later incorporated into EU acquis as successive 'Dublin' regulations performed this role (Peers et al. 2015). Following Brexit, the UK no longer had access to this means of intra-European transfer of asylum seekers from January 2021.

This further exacerbated the current challenges of managing the backlog in the asylum system, but in the UK's case, this only concerned a few hundred people who were actually transferred each year (Gower 2020).

The Rwanda plan marks a clear departure from these two approaches to enforced removals from the UK. Rather than removal directly to their country of citizenship or to a country they travelled through, individuals will be forcibly removed to a country where they have never been and they have no connections or contacts. The UK government cited precedents from the Australian government's policy of offshoring asylum processing as well as EU and UNHCR schemes to remove certain asylum seekers to Rwanda (Gower et al. 2022), but neither of these schemes is directly comparable to the UK's plans. The Australian scheme, most recently introduced in 2012 as a more stringent version of the 'Pacific solution' first implemented in 2001, involved the transfer of individuals who were intercepted in boats directly to detention centres in either Nauru or Papua New Guinea. In contrast to the UK proposition, which would transfer legal responsibility to Rwanda, the asylum claims were examined under Australian law (Matera et al. 2023). The scheme began to wind down in 2021 in the face of very widespread criticism of the Australian government for human rights abuses and massive costs in relation to the 3000 or so people who were actually transferred (Aitchison 2022; Gleeson and Yacoub 2021). Similarly, the 'Emergency Transit Mechanism' introduced by the EU and UNHCR also differs significantly from the UK-Rwanda plan. It aims to transfer refugees and asylum seekers from Libya to Niger (from 2017) and to Rwanda (from 2019), targeting individuals who have not reached European territory (Lambert 2021). The only directly comparable precedents to the UK-Rwanda plan, such as similar proposals to remove asylum seekers from Denmark and Israel to Rwanda, are equally mired in legal challenges (Denmark) (European Council on Refugees and Exiles (ECRE) 2022) or lacked any clear legislative basis (Israel) (Shoham et al. 2018).

The agreement itself is set out in the Memorandum of Understanding (Home Office 2022), signed on 13 April 2022, in Kigali between UK Home Secretary Priti Patel and Rwanda's Minister for Foreign Affairs and International Cooperation, Vincent Biruta. It consists of 24 paragraphs, followed by a further seven paragraphs in an annexe covering data protection and reporting. The preamble recognises Rwanda's significant experience hosting refugees and references the UK's 'long and proud history of providing protection'. The opening lines state the intention to 'strengthen shared international commitments on the protection of refugees and migrants'. However, the more fundamental motivation behind the MOU is explained a few lines further down in the preamble 'the prevention and combating of illegally facilitated and unlawful cross-border migration by establishing a bilateral migration partnership'. In contrast to other agreements that the UK cites as precedents, it is clear that this is not outsourcing but an abdication of responsibility to examine selected asylum claims. Once in Rwanda, individuals may claim asylum or not, but their fate is to be determined by Rwanda's legal system—in accordance with the Refugee Convention (para 2.1). The UK has already paid Rwanda £120 million under the MOU, and will likely pay more (Gower et al. 2022).

The most substantial and sustained critique of the MOU has come from UNHCR. UNHCR's initial analysis is measured but sets out a very clear basis for concern with the arrangements set out in the MOU (UNHCR 2022). The basis for the transfer arrangement is questioned since 'the primary responsibility to provide protection rests with the State where asylum is sought' (UNHCR 2022, para. 5). The review is clear that 'transfer arrangements would not be appropriate where they represent an attempt, in whole or in part, by a 1951 Convention State party to divest itself of responsibility' (UNHCR 2022, para. 9). UNHCR is concerned that the arrangement is explicitly not legally enforceable, cautions that it may undermine Rwanda's asylum system and lacks sufficient guarantees for asylum seekers concerned. It concludes that the arrangement 'is incompatible with the letter and spirit of the 1951 Convention' (UNHCR 2022, para. 26). Other areas of criticism have since

been documented, such as the mental health impact of the plan on those asylum seekers currently waiting for a decision (Smith et al. 2023).

The Rwanda deal is an attempt to address the significant backlog of undecided claims that have arisen in UK asylum policy since 2020. Like so much else in government, this situation could have been avoided simply by achieving the minimum standards of an efficient bureaucracy. Since 2010, when the current government took power, asylum applications have outnumbered decisions, so the current backlog of 160,000 people waiting in limbo has been building for some time. Keeping people waiting in the asylum system for years violates the basic principles of maintaining human dignity that the government has committed to. Yet the political narrative is driven more easily by a new idea, and this is the context in which the Rwanda plan was negotiated. Goodwin-Gill (2023) has pointed out how unusual it is for the UK government to go to the trouble of publishing non-legally binding MOUs, such as this one. Communication of this nature suggests that the government realises that the value of the MOU is to be found in the publicity that it generates amongst the UK voting public rather than the impact on reducing the backlog of outstanding asylum decisions.

The high-profile court cases may even have served this interest by further boosting publicity for the policy and allowing a government that had been in power for 12 years to present itself as somehow anti-establishment. The initial judgement (EWHC 3230 2022) on the eight separate challenges to the Rwanda policy, published on 19th December 2022, was broadly in the government's favour. The separate cases, heard together, involved 11 individuals who had been identified for removal to Rwanda and four organisations who challenged the policy on behalf of others. The Home Office had argued that three of the four organisations did not have the necessary standing to pursue challenges since they were not directly affected by the outcome. The court agreed, meaning that these organisations are unable to pursue any appeal. The judgement reduced the multiple challenges in each of these cases down to 12 'generic' claims (para. 39), all of which it eventually rejected. This rejection is clear early on in the 139-page judgement. The judgement recognises the public interest in the case and the range of questions that the relocation of asylum seekers to a third country inevitably raises but is then clear in its remit (para. 5). 'But those matters are not for the court. The role of the court is only to ensure that the law is properly understood and observed and that the rights guaranteed by Parliament are respected.' Since Parliament had recently legislated, in the Nationality and Borders Act 2022, to allow enforced removals to Rwanda, the court went on to find that those powers were indeed lawful. For a number of individuals, the court found that in relation to the particular circumstances of their claim, their challenges could be upheld, and they were given permission for judicial review. On all of the challenges to the policy as a whole, however, the court found that the government did indeed have the power to act to remove individuals to Rwanda.

The Court of Appeal judgement, on 29th June 2023, is less complicated (EWCA Civ 745 2023). The UNHCR intervened more formally in this case, providing evidence that Rwanda could not be considered a safe country. The court agreed that it was not safe to deport asylum seekers to Rwanda due to problems with the Rwandan asylum system. However, the court did not agree with the principle that asylum seekers could be deported to a country where they had not been for the purposes of examining their asylum claim. The case is likely to go to the Supreme Court.

The very widespread criticism that the deal has received, its extremely high cost, and the fact that the initial plan to forcibly remove only 200 individuals of the more than 160,000 awaiting a decision suggest that it is unlikely to meet its stated objectives. The real value for the UK government is to be found in a performance of control. This is important and helps to reinforce our central argument that the historical precedent to the Rwanda plan is to be found less in contemporary (post-1970) deportation policy and much more obviously in colonial practices of directed migration and refugee relocation or dispersal. It is to this subject that we now turn.

### 3. Colonial Practices of Forced Relocation

The story of migration in the British Empire is the subject of many different histories told from several angles of vision encompassing internal, intraregional, and global migrations (Harper and Constantine 2010). These histories span the various forms of British rule in settler colonial states and British-administered colonies without significant settlement, governed under diverse arrangements and comprising colonies, protectorates, mandates, and dependent territories. Within this history, extensive scholarship has looked at a series of forced relocations of different populations in different contexts and times. From the establishment of the British colonies in sixteenth and seventeenth-century America to the twentieth-century transfer and redistribution of refugees throughout the length and breadth of the British colonies, numerous instances illustrate how the forcible relocation of people was foundational to the establishment of colonialism and colonial borders over the long duration of British rule over a vast geography (Seeley 2021; Wolfe 2006). It is impossible to discuss every case of forced relocation in the British Empire or to linger over the nuances of these histories that also frequently encompassed elements of migrant choice and agency in the process of relocation as well as resistance to relocation and insistence on the right to remain (Seeley 2021). However, we aim to show that while the Rwanda plan represents a departure from the UK's post-1970s deportation policy, it does not reflect a contemporary invention. Through a synthesis of the scholarship on relocation in the Empire, we draw attention to some pivotal precedents to establish that the concept of 'offshoring' migrants and refugees, notably those considered to be undesirable and at the bottom of racialised hierarchies, is part of an extended history of colonial forms of thinking and practice on the physical location of racialised peoples.

The Atlantic slave trade (c.1500–c.1900) is the most potent reminder of the forced relocation of people, mainly enslaved Black Africans to the Americas to labour on the plantations. Settler colonialism relied on the forced migration of the enslaved both overseas and within colonial America (Pargas 2014). In addition to the economic exploitation of the enslaved, Indian Removal was also crucial to the annexation of land and the expansion and establishment of the United States (Seeley 2021; Littlefield and Parins 2011). Dispossession, removal, and eviction of indigenous populations was a characteristic feature of white settler colonies (Knafla and Westra 2011). Within Britain itself, Scottish and English planters settled the Ulster plantations from the early 1600s, resulting in the usurpation of the land from the Irish. This played an essential role in the English conquest of Ireland (Ciardha and Siochrú 2012). In the Scottish Highlands and Islands, the English forced evictions of Scottish tenant farmers from the latter half of the eighteenth century to the mid-nineteenth century. The legacies of these clearances for Highlanders included impoverishment, being forced onto barren land, death, and emigration to other parts of the Empire (Richards [1982] 2023).

Another prominent form of colonial forcible relocation in the eighteenth and nineteenth centuries was transportation. Transportation was a practice followed by several European empires and 'was a means of punishment, deterrence, population management and, through the expropriation of convict labour, of occupying and settling distant frontiers. Convicts travelled multi-directionally, shipped outwards from Europe and other metropolitan centres, within nations, and between colonies and the so-called peripheries of empires and polities' (Anderson 2018). The forced transportation of English, Irish, Welsh, and Scottish convicts to overseas colonies at first in North America and the Caribbean, and then, after the American Revolution, to Australia (Kercher 2003), was both a form of punishment meted out to them as well as a device to mobilise labour and assisted the expansion of British colonial power in the seventeenth and eighteenth centuries (Maxwell-Stewart 2018). Contemporary attempts to intercept and turn back boats with refugees on the Channel are linked to the 'enduring colonial fantasy' of maritime "offshoring"' and colonial bordering and echo Britain's tradition of sending racialised and 'undesirable' populations to the colonies (Davies et al. 2021).

An outstanding illustration of the British relocation of refugees in the eighteenth century is that of Black exiles of the American Revolution who fought alongside the British

whilst simultaneously fighting enslavement. They ended up in the 'howling wilderness, at opposite ends of the globe' (Pybus 2006). A community of slave fugitives found themselves first in New York and then in Nova Scotia before approximately three thousand Black men, women, and children were evacuated and made a difficult journey to Sierra Leone in West Africa in 1792. They were promised freedom, self-determination, and land grants here, but these promises amounted to little (Pybus 2006). Black refugees also found themselves in the penal colony of Botany Bay in Australia via London. Black refugees had arrived in London destitute. They received a hostile and racist reception and no compensation for their services to the British. They made clear to the Loyalist Claims Commission that they had not freely chosen England, where they suffered and had been forced to leave their wives and children in America. As Cassandra Pybus has shown, penniless Black refugees forced into theft were among the first convicts transported to Australia (Pybus 2006).

In the eighteenth and nineteenth centuries, in colonies such as India, the British colonial state attempted to limit some forms of mobility while promoting others (Chatterji and Washbrook 2013). Wherever the British built new port towns and founded new colonial states, there was a significant demand for infrastructure and qualified workers. Unable to provide this for themselves since it was expensive to maintain Europeans in the tropics and to keep them alive, the colonial Indian state facilitated the migration of clerks, accountants, soldiers, and policemen to work in Singapore, Burma, and Malaya; Sri Lanka; east and south Africa; Aden and the Persian Gulf; and Hong Kong and Shanghai, in large numbers (Washbrook 2013; Metcalf 2008). 'Wherever the leading projects of colonial capitalism—in mines, plantations and construction—ran into problems of labour-shortage, whether from under-population or resistance,' the shortages were answered by the recruitment of workers largely through the system of indenture (Washbrook 2013).

On the other hand, a seemingly contrary but related focus of British policy, particularly in the nineteenth century, was to immobilise certain communities by confining them to specific locales. This could involve, for example, the sedentarisation of people on the land (Chatterji and Washbrook 2013) or the criminalisation of itinerant groups. Across Europe and India, several mobile groups were categorised as 'criminal' (Dragomir 2019). 'They comprised the loafers, drifters, strollers, conmen, charlatans, conjurers, acrobats, jugglers, wanderlusts, mendicants, ascetics, floaters, rovers, prostitutes, flunkeys, gypsies, vagrants, peripatetic, itinerants, vagabonds, fugitives, listless, the indolent and the nomads' (Rana 2011). These groups occupied the 'margins of the margins', whose 'mobile and unregulated' lifestyle colonial authorities perceived as a threat to their established order (Rana 2011). 'Criminal tribes' were relocated and then confined to certain areas (Schwarz 2010), engendering the forced immobility of mobile groups identified as undesirable and were sometimes forcibly settled in ghettos or reformatory compounds (Schwarz 2010). South Africa's Pegging Act[4] and Ghetto Act,[5] the Indian Foreigners Acts, the Criminal Tribes Acts[6] and the Hur Acts of Sindh[7] exemplified legislation that concerned the restriction of certain groups of people to specific areas. Although it is beyond the scope of this article to discuss the history of segregation and apartheid in any detail, the forcible relocation and restriction of Black South Africans into separate 'homelands' under European colonial rule is one of the most conspicuous demonstrations of how forced relocation and immobility went hand-in-hand.

As the above discussion illustrates, over the British imperial chessboard that spanned vast territories, the steering of migration in distinct directions and the simultaneous desire to enforce immobility over migrant and itinerant groups were enduring features. Behind colonial techniques of rule was the notion of the world, and the Empire, as a unified but hierarchical whole in which race became central to the organisation of power (Chatterjee 1993). Moving people in particular directions, especially those who were situated on the bottom rungs of colonial categories of race to parts of the Empire outside of the UK, acquired a particular salience in the twentieth century.

In contextualising the Rwanda deportation scheme, it would be apposite to consider the colonial 'politics of dispersal' (Cosemans 2022; Shahani 2021), relocation, and repa-

triation in the twentieth century. The contemporary politics of dispersal emerge from several different strands of colonial history, some of which we have discussed above. We focus on two here: the colonial direction of migrant labour, either to or away from specific territories in the early twentieth century and the planned distribution of refugees during the interwar years to colonised territories. Significantly, the UK-Rwanda scheme is touted as an economic development programme. As we will show below, the UK sometimes justified the dispersal of refugees to the colonies on the grounds that the relocation of refugees would result in 'development' and economic progress in the receiving territories.

The system of indenture, greatly expanded after the abolition of slavery, involved a contract and an element of 'consent' (including a promise of 'return' after five years of labour because the colonial state wanted to ensure that indentured labour migration was seen as 'free' as opposed to the slave trade (Mongia 2018). The British introduced the regulation of migration at first to *facilitate* the movement of indentured labour (Mongia 2018) in specific directions. Historians have debated what constituted 'consent' and the meaning of 'consent' for indentured labourers transported to places and working conditions of which they were ignorant (Koya 2020). But for our purposes, it is pertinent to note that a chimerical notion of consent exists in the UK-Rwanda plan. The Memorandum of Understanding states that asylum seekers will be free to leave Rwanda if they choose. If asylum seekers deported to Rwanda from the UK do not wish to stay, they will be sent back to their country of origin or to a 'country in which they have the right to reside.'[8] (Home Office 2022, para10.4) However, it is not clear how they will make these journeys out of Rwanda.

From the early twentieth-century colonial legislation to 'regulate' migration would proliferate and move away from what Radhika Mongia (2018) calls a 'logic of facilitation' to encompass a 'logic of constraint'. Although British imperial subjecthood was theoretically universal, colonised subjects were increasingly constrained by racialised migration controls in the United States and the British Empire. These controls emerged consequent to pressure on the metropole from the white settler colonies, who imagined their emerging national sovereignties in terms of white supremacy. The white settler colonies perceived indentured and free Asian migrant labour as a threat to their political and economic dominance. The Colonial Office in London tried to stave off restrictive legislation that would destabilise Britain's relationship with Asian powers like Japan and cause unrest in colonies like India. But it ultimately had to give way, allowing restrictive immigration laws targeted at non-white emigration to the settler colonies. (Atkinson 2016; Martens 2018). In certain regions, like the Transvaal, some British officials and settler capitalists argued for the continued exploitation of Chinese labourers who were hired as temporary expedients. They were repatriated as soon as they had completed the tasks for which they had been imported so that they would not form permanent settlements (Atkinson 2016). The desire to turn the supply of Asian labour on and off at will and to prevent Asian settlement in the white settler colonies was characteristic of what David C. Atkinson calls a commitment to 'white mobility and nonwhite immobility' and the 'impermeability of colonial borders' (Atkinson 2016). As we show below, this colonial characteristic of 'regulating' migration to propel it in specific directions while also promoting immobility is one we see well into the twentieth and twenty-first centuries, especially when it came to the relocation of refugees in the interwar years.

The Foreign Office had repeatedly pressed the Colonial Office in the early 1930s, in response to a request from the League of Nations, to investigate the possibility of allowing some 10,000 Assyrian refugees who had been forced to flee Iraq to settle in Cyprus, East Africa, Ceylon, Mauritius, Seychelles, Tanganyika, Northern Rhodesia, British Guiana, Nyasaland, or Uganda. The League followed this with another request in 1935 to accommodate Jewish, Assyrian, Turkish, Armenian, Russian, and Saar refugees in the colonies (Newman 2019). These schemes did not come to fruition, but during the Second World War, Britain succeeded in overseeing the relocation of large groups of people to areas with which they might have no previous connection. The relocated refugees were those construed as 'subaltern whites' (Fischer-Tiné 2009) or those who occupied the 'edges of

whiteness' (Lingelbach 2020). The dispersal of poorer refugees and refugees considered of inferior status to colonised territories in the Caribbean, Africa, and India (Lingelbach 2020; Shahani 2021) involved the continuation of a practice of relocating prisoners of war and refugees across the Empire during the Boer War and World War One (Kennedy 2016). The absence of a legal definition for 'refugee' in the interwar years meant that the term was used flexibly to describe different groups, and the categories a displaced person occupied could shift (Kapoor 2022). The Colonial Office was against plans to relocate the war-displaced to the colonies, as were white settlers and administrators on the ground (Lingelbach 2020; Newman 2019; Bhatti and Voigt 1999). Nonetheless, several colonies, including colonies in the Caribbean, Africa, British India and princely India, and Australia, served as "holding" locations during the war for diverse populations, including Jewish refugees, Stalin's freed Polish prisoners, Czech, Hungarian, Italian, and Bulgarian prisoners of war, political prisoners, dissidents, and Maltese and Cypriot subjects of the British Empire. These groups could either be confined to specific camps or interned as 'enemy aliens'. Indeed, refugees could also occupy the category of 'enemy aliens' (Gilman and Gilman 1980; Bhatti and Voigt 1999).

Dispersal acquired further compelling urgency after the war when decolonisation became a tangible prospect, and India won independence in 1947. The newly independent country was reluctant to continue hosting the refugees of Europe's war to which it had been signed up without consultation. One of the reasons for the Indian government's attitude to the European displaced was due to India's own experience of 'transnational whiteness'. Although displaced white Europeans in the colonies occupied a subordinate class of whiteness, the British treated them more favourably and accorded them more humanitarian assistance than displaced British Indians, approximately half a million of whom became refugees during the Second World War but were not necessarily recognised as such (Kapoor 2022). Moreover, India was soon gripped, along with Pakistan, by its own refugee crises of mammoth proportions of approximately 15 million displaced, generated by partition in 1947. At this time, the UK did not want the displaced populations it had interned in India to so much as transit through the UK to return home or to go to other countries. The Foreign Office went so far as to refuse to accept that the British had interned myriad groups of European displaced persons in India until the Commonwealth Relations Office produced 'chapter and verse' evidence (Shahani 2021). The Foreign Office then maintained that the solution was to simply shift the 'problem' of these displaced persons elsewhere, to geographically distant locations in still-existing African and Caribbean colonies (Shahani 2021)[9].

The Caribbean, where the British had transported enslaved and indentured labour, was also a place Whitehall envisaged as a home for refugees it did not want in the UK, despite ongoing resistance to the idea from the Colonial Office.[10] Jewish communities had settled in parts of the Caribbean since the seventeenth century. Their presence increased in the British West Indies in the 1920s and then as they began to flee Nazi persecution, particularly when they were unable to enter the UK or the United States (Newman 2019), but they were not always welcome. Nevertheless, in 1948, the Secretary of State for the Colonies presented to the UK Parliament the Report of the British Guiana and the British Honduras Commission under the Chairmanship of Sir Geoffrey Evans.[11] The purpose of the Evans Commission was to investigate and report to the Secretary of State for the Colonies on the feasibility of transferring and settling in British Guiana (Guyana) and British Honduras (Belize) 'surplus' Jewish refugees and 'surplus' populations of other West Indian Islands. The British government had repeatedly considered these regions in the face of antagonism from the Colonial Office and local governors and internal reservations about the feasibility of these schemes as it did not want to host these Jewish refugees in the UK. Furthermore, it wanted to divert attention from plans to restrict emigration to Palestine (Newman 2019). As the disagreements between the Colonial Office and local governors with the British government on the resettlement of Jewish refugees in the Caribbean demonstrate, while various officials within the government could express qualms about

these relocation schemes, the belief that the colonies would ultimately have to serve as the place for unwanted refugees persisted (Newman 2019).[12]

The British frequently justified colonial rule on the basis that the colonies were ill-suited to govern themselves. The purpose of colonial rule was to help the colonised grow into the qualities they needed for self-government (Coupland 1948). The idea behind 'developmental colonialism' was that white colonisers would guide the colonised to better standards of living (Lingelbach 2020). In the decolonising world of the 1940s, in some cases, refugees were cast as the agents of this development (but sometimes as competitors to local populations). A Royal Commission had reported shortly before the Second World War that agricultural development in British Guiana was unlikely (Cumper 1949/1950). Contrary to the findings of that Report, the Evans Commission suggested that land in British Guiana was fertile and framed the refugees as pioneers who would develop British Guiana and Honduras and stand to benefit themselves. It proposed that the two colonies absorb one hundred thousand immigrants over the next ten years, who could work mainly as plantation owners, agricultural labourers, and smallholders.[13] Similar policies to move significantly poorer and caste-oppressed refugees to remote locations and 'empty' corners considered too barren and inhospitable for others proved popular in postcolonial India. The new government of independent India cast partition refugees it wished to disperse to far corners in the mould of nation-builders and as vehicles of economic planning and development (Chatterji 2007; Sen 2018). Similarly, the UK-Rwanda scheme is officially known as the UK and Rwanda Migration and Economic Development Partnership. Doris Uwicyeza, lead legal negotiator for the partnership, has implied that the welfare of particularly vulnerable refugees would not be a concern, as Rwanda would receive 'substantial investment that would benefit all migrants and all refugees in Rwanda.'[14] The Rwanda scheme promises 'investment' both for the refugees and Rwanda, echoing official colonial schemes that legitimised the necessity of dispersing refugees to territories considered marginal or inferior with the language of economic 'development'. Indeed, the UK Home Office's 'fact sheet' on the Rwanda scheme promised that it would 'enhance economic prosperity in the region by investing in upskilling, development and projects which will benefit both migrants and their hosts' (Gower et al. 2022).

The 1951 Convention on the Status of Refugees defined refugees as Europeans displaced before 1951, thus leaving out vast swathes of colonised, formerly colonised, and racialised asylum seekers from regions also facing massive refugee crises (Oberoi 2006, Mayblin 2017; Krause 2021). The racialised territorial and temporal restrictions on the Convention were lifted in 1967. However, merely a year later, in 1968, the UK enacted the Commonwealth Immigrants Act 1968, popularly known as the Kenyan Asians Act, which sought to block the entry to Britain of East African Asians. The UK had already started to erode the right of commonwealth citizens, as opposed to white immigrants, to enter the UK with the passage of the Commonwealth Immigrants Act 1962 (Paul 1997) when East African Asians began to leave Africa in response to Africanisation policies that disenfranchised them. Their exodus peaked in 1972 when Idi Amin expelled them from Uganda. Many of the East African Asians were British citizens, and the Act prevented Britain's own citizens from moving to their country of nationality. In *East African Asians v. The United Kingdom*, a group of East African Asians challenged this legislation as discriminatory before the European Commission of Human Rights. [15] The claimants were successful, but the Commission found that some were British Protected Persons (BPPs) with no right of abode in the UK. The Commission held that the exclusion of BPPs from the UK was permissible, and they became stateless refugees. As Sara Cosemans has shown, the British government, with the assistance of UNHCR, dispersed BPPs and other stateless East African Asians to at least twenty-five international destinations (Cosemans 2022). The history of the East African Asians' case, we suggest, also provides a pertinent historical background to the direction that the UK's approach to postcolonial citizenship and asylum has taken with the UK-Rwanda scheme in juxtaposing regimes of non-entrée with regimes

of dispersal or relocation. As we have argued, these practices of facilitating movement in a pre-determined direction or impeding it were enmeshed in the colonial past.

## 4. Conclusions: Deportation as Colonial Power Play

The current UK government's continued pursuit of the UK-Rwanda Migration and Economic Development Partnership highlights a willingness to prioritise the Prime Minister's policy pledge to 'Stop the Boats' above the international reputation of the UK and, perhaps less surprisingly, the welfare of refugee groups. The policy is undoubtedly popular amongst certain members of the electorate, even before any asylum seekers have been deported to Rwanda, but it is still a gamble. Those who feel strongly and positively about the policy are a small minority and are probably outnumbered by those who feel strongly and negatively. For it to have any measurable effect in reducing undocumented migration across the English Channel, the policy would also have to grow so far beyond the initially envisaged 200 people that it becomes unaffordable. Still, governments betting on anti-migrant sentiment for short-term electoral gain is hardly news, certainly in Europe or North America.

The more novel question, which this paper has pursued in detail, is the nature of the colonial antecedents to this policy. In concluding this paper, we consider why they matter. The colonial echoes of this policy do not mean that the Rwandan government did not enter into the policy voluntarily. There are several obvious advantages for the Rwandan government, even on top of the £120 million initial payment. The President, Paul Kagame, has long faced charges of human rights abuses, and it has not hurt his international image to have UK government ministers regularly describing his country in glowing terms (Akanga and Hughes 2022). Yet Rwanda is far from a democracy (Freedom House labelled it 'not free' in both its 2022 and 2023 annual reports), so the agreement is with a small, unrepresentative group of Rwanda political elite. Even if Rwanda were more democratic, a payment of £120 million, not a particularly substantial sum by UK government standards, would certainly have had a force that bordered on coercion. The opposition Democratic Green Party of Rwanda came out strongly against the deal, arguing that the UK government should not shift its obligations just because it had 'the money and influence to enforce their will' (Suuna and Burke 2022).

The colonial connections of this current policy matter because they reinforce a system in which these kinds of asymmetric power relations are normalised. In turn, this reproduces and reinforces relationships between state institutions in different countries and between state institutions and disempowered migrants that are similarly hierarchical. Consciously or unconsciously, migrants affected by this policy are defined in racialised terms, further reinforcing those ethnic boundaries. For individuals, this policy goes well beyond those threatened with deportation. It shifts the boundaries between belonging and not belonging, placing barriers to full participation in society (Collyer et al. 2020). The normalisation of colonial relations between states further exacerbates the difficulties of finding a more sustainable solution to what has become a global crisis of displacement. The vast majority of displaced people in the world are hosted in formerly colonised countries. Policies that shift the obligation to care for even small numbers of displaced people from wealthy to poorer countries reinforce an idea that they are 'surplus', a problem to be passed on and an obligation that can easily be avoided. In turn, this serves to legitimise the current global distribution of displaced people, reinforcing the notion that the many millions hosted are surplus and, as such, can be abandoned in terrible conditions in desperately poor countries.

Britain's colonial practices, examined in the second part of this paper, highlighted how surplus populations could simply be shifted around the Empire. Nostalgia for the Empire has become more acceptable in the UK since Brexit and correlates strongly with the anti-migrant sentiment that the current UK government is counting on for electoral success. Yet the relevant period of Britain's colonial history is instructive here. Although forcible population movement is a characteristic of most colonial regimes, unwanted refugees were moved around the British Empire in larger numbers after the Second World War at the

very end of the Empire. Then, as now, the colonial response highlighted not the continued power to influence global events but rather the lack of that power. In the late 1940s, Britain was indebted and damaged by the war but remained able to shift problems it did not have the financial capacity to respond to. In 2022, Britain did not turn first to the policy solution of twenty years earlier and focused predominantly on increasing decision-making and subsequently increasing forced removals. Cost is certainly a factor in the more straitened economic times of 2022, and the high per person cost of forced removals mitigated against their widespread use, though there are others, such as a high turnover of relevant staff. There is also a strongly performative element to the Rwanda scheme, and the attention it has attracted must be viewed as deliberate; simply enhancing existing bureaucratic practices would not have conveyed the same message.

The Rwanda scheme is best understood as a colonial power play in all senses of the term. Most analysis agrees that whatever the policy is, it cannot be interpreted as a serious attempt to resolve the long-standing political challenge of asylum in the UK since, even if it were to be enacted, it would not make a serious contribution to this objective. In the most immediate sense, it is a play for power, an attempt by the current UK government to shore up the electoral support that has deserted it in recent years. It is also a play in the sense of a performance. It appears to be a dynamic, original response to a well-recognised policy priority. From the outset, it was clear that the policy would be opposed by a civil society committed to the defence of the principle of asylum and the rights of refugees and most likely involve the UK courts and the European Court of Human Rights as well. The further electoral gamble was that a sufficiently large number of people would view a public battle with any of these groups sympathetically to carry significant electoral weight.

These are the types of calculations made by any democratically elected government, although much political analysis of the current government has come to assume a particular level of cynicism in these calculations. The key difference in this policy is the specific colonial references analysed here. As we have shown, it echoes a set of practices common throughout Britain's colonial period but especially widely used, and envisaged, for the relocation of refugees towards the end of the Empire. The ultimate impact of the policy, even if it fails in its explicit objectives, will relate to the legitimation of notions of surplus population. This concerns both who they are, in specifically racialised terms, and where they are. The damage to the international system may, therefore, go beyond the immediate effects of undermining the 1951 Convention, as UNHCR has argued, serious though that is. Ultimately, the Rwanda scheme and other colonial-influenced migration and asylum policies provide legitimacy to the current vastly unequal global distribution of refugees, confirming the justification of expulsion.

**Author Contributions:** Conceptualization: M.C. and U.S.; Original Draft preparation: M.C. and U.S. All authors have read and agreed to the published version of the manuscript.

**Funding:** This research was supported by the British Academy (grant number CRFG\IC4/100285).

**Institutional Review Board Statement:** Not applicable.

**Informed Consent Statement:** Not applicable.

**Data Availability Statement:** Not applicable.

**Acknowledgments:** We are grateful to Anne Irfan, who directed the project.

**Conflicts of Interest:** The authors declare no conflict of interest. The funders had no role in the design of the study; in the collection, analyses, or interpretation of data; in the writing of the manuscript; or in the decision to publish the results.

## Notes

1    In the UK, there is a legal distinction between 'enforced removals', which involves the removal of any foreign national from the country against their will and 'deportations', which applies specifically to foreign national prisoners on release from prison. In the wider literature on this subject, 'deportation' is much more common and is used to refer to any forcible removal. We use 'enforced removal' in reference to UK statistics and practice but use the term 'deportation' as an overlapping but broader term to refer to international practice.

2    The selected primary sources are primarily from the National Archives in the UK, which houses the files dealing with government policy on the relocation of refugees.

3    This figure appears as 'Persons removed as a result of enforcement action' in the annual *Control of Immigration* reports (Home Office 2002, 2006, 2010) and online statistics since 2013 (Home Office 2013–2023). Before 1992, asylum applicants were not separated in these statistics. Several different figures appear in different reports as figures are typically revised in subsequent years. Data for Figure 1 is taken from the latest reports in which the year's data is available.

4    Officially known as The Trading and Occupation of Land (Transvaal and Natal) Restriction Act 1943, it prevented Indians from buying property in certain areas of Durban for occupation and investment.

5    The Asiatic Land Tenure and Indian Representation Act 1946, named the 'Ghetto Act' by Indians in South Africa, extended the provisions of the Pegging Act to cover the whole of Natal and Transvaal. The Act also prohibited Asians from owning or occupying property without a permit when such property had not been owned or occupied by Asians before 1946.

6    The Criminal Tribes Act, 1924, (VI of 1924). The British government first introduced the Criminal Tribes Act in 1871. It went through several amendments until a consolidated version was enacted in 1924.

7    Sind Suppression of Hur-Outrages Act 1942 and Sind Hurs Detention Act 1947.

8    'UK to take 'most vulnerable' refugees from Rwanda under deal, says scheme's negotiator,' https://www.lbc.co.uk/radio/presenters/tom-swarbrick/uk-take-vulnerable-refugees-from-rwanda-deal-scheme/, accessed on 29 July 2023.

9    'Proposal to settle Maltese refugees in India in British-occupied territories in Africa,' FO 1015/52, 1015/53 & 1015/54, The National Archives; 'Repatriation of Bulgarian and Hungarian internees from India', 1947; 'Removal of Polish refugees from India', 1946-47. HO 213/1190, The National Archives.

10    Settlement of Jews and West Indians in British Guiana and British Honduras. Code 48, file 1386, FO 371/72093A, The National Archives.

11    An economic botanist and former principal of the Imperial College of Tropical Agriculture.

12    Report of the British Guiana and British Honduras Settlement Commission, 1948, (London: H.M. Stationery Office).

13    Ibid.

14    See note 8.

15    *East African Asians v United Kingdom* [1973] 3 EHRR 76.

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
