# Peer review of "Offshoring Refugees: Colonial Echoes of the UK-Rwanda Migration and Economic Development Partnership"

_socsci, doi:10.3390/socsci12080451_

Round 1
Reviewer 1 Report
I found this an excellent paper which demonstrates clearly the colonial echoes in the UK's current policy of deporting refugees to Rwanda, and the importance of the way in which these colonially inspired practices are being normalised through this policy
Just a couple of small questions
1. You talk about "clandestine" small boat arrivals? Is this a perjorative term? Or one that is used generally in scholarship on the issue? I'm just asking because of all the discussion about use of "illegality" etc in regard to arrivals.
2. In this sentence: Despite this wide-ranging opposition to 142 enforced removals, deportation performs a necessary role in what William Walters (2002) 143 has called the ‘international police’ of populations. The enforced removal of an individual 144 from a country where they are found to be illegally resident to their country of citizenship 145 reinforces the bond between citizenship and territory, which is essential to the interna-146 tional management of populations within the international state system (Hindess, 2000).
You seem to be saying that deportation is "necessary"? perhaps I'm misreading or it needs to be better phrased?
3. You say that the UK's withdrawal from the EU (and thus from Dublin) " "further exacerbated the current challenges of managing the backlog in 170 the asylum system,"
But isn't there a strong argument that has made the the UK government has more or less deliberately let this backlog build up and indeed that it might be a deliberate tactic to gain support for their Rwanda policy?
Author Response
Many thanks for this helpful and positive review
- You talk about "clandestine" small boat arrivals? Is this a perjorative term? Or one that is used generally in scholarship on the issue? I'm just asking because of all the discussion about use of "illegality" etc in regard to arrivals.
Thanks for highlighting this inconsistency. 'clandestine' is used to refer to undetected arrivals - it has no pejorative connotations. In the case of small boats, many arrivals are not actually 'clandestine' since they are detected. We have therefore changed first one to ‘irregular’ while maintaining the second (referring to arrivals by lorry, which were often not detected).
- In this sentence: Despite this wide-ranging opposition to 142 enforced removals, deportation performs a necessary role in what William Walters (2002) 143 has called the ‘international police’ of populations. The enforced removal of an individual 144 from a country where they are found to be illegally resident to their country of citizenship 145 reinforces the bond between citizenship and territory, which is essential to the interna-146 tional management of populations within the international state system (Hindess, 2000).
You seem to be saying that deportation is "necessary"? perhaps I'm misreading or it needs to be better phrased?
We’ve rephrased this to remove the ambiguity – the argument that we’re making (following others who are cited) is that deportation is an inevitable result of an international system of states in which citizenship is associated with states. This is a necessary point since it explains why the majority of deportations involve the forced removal of individuals back to their country of citizenship – the Rwanda scheme differs from this.
- You say that the UK's withdrawal from the EU (and thus from Dublin) " "further exacerbated the current challenges of managing the backlog in 170 the asylum system,"
But isn't there a strong argument that has made the the UK government has more or less deliberately let this backlog build up and indeed that it might be a deliberate tactic to gain support for their Rwanda policy?
This certainly appears possible, though mismanagement is usually a more convincing explanation than conspiracy and in the absence of any evidence that this was a deliberate tactic we don't think we can substantiate this kind of argument.
Reviewer 2 Report
The article tackles an important topic - the colonial genealogy of British schemes such as the recent "Rwanda plan".The structure reads well, however there is a lack of profundization in several concepts/categories.
Here go my suggestions:
1) Reconsider the initial division between the "physical" and the "political", since according to postcolonial, feminists, Foucauldian and other theories, the body is indeed one of the major sites of politics; maybe what the author are looking for is more a physical/symbolic duality?
2) consider using "rejected" instead of "failed" asylum seekers, in order not to reify the language of the asylum regime.
3) at some point the authors refer to the 2012 Australian scheme. Are they're referring to the Pacific Solution? If so that traces back to the early 2000s, after the so called "Tampa accident"; the reference to this needs to be more specific.
4) the difference the authors state between the Australian/Pacific and UK/Rwandan schemes is not fully spell out. While they state that the UK declines any responsibility, Australia has let people in detention for several years (Behrouz Boochani's book would be worth quoting on the matter). By this perspective, the matter of responsibility seems only a formality, so the difference should be stated more clearly.
5) reconsider reifying the idea of a "crisis" of the asylum system (p. 5). the system was always already in "crisis" from the perspective of asylum seekers.
6) the authors conclude that the true value/goal of the Rwandan scheme is a performance of control - but isn't that true of most border control? Most border studies demonstrate how borders are never sealed but rather a spectacle of control (De Genova) or a matter of filtering and differential inclusion (Mezzadra).
7) The authors exclude from their analysis any economic aspect - sure there any many private companies/contractors that benefit from this kind of deportation agreements?
Sentences to revise:
1) check the last sentence of the firs paragraph, it does not read well grammatically.
2) check the first paragraph of p.21, it reads awkward.
3) "getting the basics right" (end of p. 5) or "it's in no one's interest" (p. 6) read too informal
Author Response
Many thanks for this really helpful review. We're particularly grateful for highlighting the lack of clarity in the language and potential ambiguities in the expressions that we've used. We respond to each of the points raised below.
1) Reconsider the initial division between the "physical" and the "political", since according to postcolonial, feminists, Foucauldian and other theories, the body is indeed one of the major sites of politics; maybe what the author are looking for is more a physical/symbolic duality?
Many thanks for this – this was the point that we were making and you’re right to highlight that the language we were using confused rather than clarified the issue. We have re-worded this, removing the distinction, which hopefully clarifies this.
2) consider using "rejected" instead of "failed" asylum seekers, in order not to reify the language of the asylum regime.
We’ve made this change.
3) at some point the authors refer to the 2012 Australian scheme. Are they're referring to the Pacific Solution? If so that traces back to the early 2000s, after the so called "Tampa accident"; the reference to this needs to be more specific.
The Pacific Solution was abandoned in 2008, but an even harsher set of penalties was introduced in 2012, so we’re just referring to the post-2012 policy here, but we’ve referenced the Pacific solution to make the link clear.
4) the difference the authors state between the Australian/Pacific and UK/Rwandan schemes is not fully spell out. While they state that the UK declines any responsibility, Australia has let people in detention for several years (Behrouz Boochani's book would be worth quoting on the matter). By this perspective, the matter of responsibility seems only a formality, so the difference should be stated more clearly.
We’ve tried to make this clearer – the question of which legal system an asylum claim is examined under is much more than a formality. As you suggest, there are similarities between the proposed UK and abandoned Australian system (the possibility of indefinite detention under both approaches is an obvious cause of tremendous suffering, as Boochani powerfully attests) but the purpose of this section is to highlight the differences in legal process, and to do so fairly concisely, so we’ve not dwelt on what the systems have in common.
5) reconsider reifying the idea of a "crisis" of the asylum system (p. 5). the system was always already in "crisis" from the perspective of asylum seekers.
We’ve changed this language.
6) the authors conclude that the true value/goal of the Rwandan scheme is a performance of control - but isn't that true of most border control? Most border studies demonstrate how borders are never sealed but rather a spectacle of control (De Genova) or a matter of filtering and differential inclusion (Mezzadra).
We totally agree this is the conclusion of most (critical) border studies work – which is why we don’t present it as a novel conclusion here, this is only a summary point at the end of the first section, it does not appear in the main conclusion to the article.
7) The authors exclude from their analysis any economic aspect - sure there any many private companies/contractors that benefit from this kind of deportation agreements?
This is definitely true and we’ve written about this elsewhere, the main points have been made by others (such as Ruben Anderson) but at this stage we don’t know what companies will implement the Rwanda scheme so direct empirical work is not possible and this isn’t the main focus of this article.
Comments on the Quality of English Language
Sentences to revise:
1) check the last sentence of the firs paragraph, it does not read well grammatically.
We’ve rephrased this sentence.
2) check the first paragraph of p.21, it reads awkward.
It’s not clear what paragraph is being referred to here as there are only 13 pages.
3) "getting the basics right" (end of p. 5) or "it's in no one's interest" (p. 6) read too informal
We’re rephrased these two phrases.
Reviewer 3 Report
This is an interesting and well-researched article that makes a strong contribution to scholarship.
I do have some suggestions, however. While noting that there are six footnoted archival sources, it did seem that large portions of the paper are without source citations. And even the sections with the footnoted archival citations may be lacking some in-text source citations.
Given the mention in the abstract that this paper is "based on extensive archival research," I expected to see more of this up front and to also to see a little more about the research process.
Was a particular archive examined? Were particular time periods or locations examined? How were these documents found/selected?
While the documents used, and the material drawn from them, is excellent, there is an element of randomness to their presentation that makes me question how "extensive" the archival work actually was. I would like to hear more about the research process and the parameters of inquiry.
In addition, the abstract repeatedly stresses the colonial connection - "Colonial Echoes" is in the title -- and one reads in the abstract about colonial policies, colonial states, colonial practices.
Yet the essay does not really engage the colonial theme until page 6, and then only for about three pages. I think the colonial aspect could be more prominently presented. The word appears 16 times before section 3 (five of these in the abstract and title), a further 23 times in section 4 (beginning on page 6), about 12 times in the concluding section (beginning on page 10), and eight times in the bibliography. Given the early emphasis on it, I had expected to encounter more about the colonial context and to do so earlier in the paper. As it is, despite the earlier promise, the attention to the colonial past really receives only about 3 1/2 pages of attention in 11 pages of text, and most of that not until the second half of the essay.
It might be worth considering a slight reorganization of some material if the colonial focus/thrust is really central to the author's conception of the piece.
The essay also leans far more heavily towards the second British empire than the first. In some ways this may be appropriate, but I think there is material that could be profitably incorporated that could strengthen the argument. If the focus is solely on the later colonial period, though, this could be better explained.
The early British experience in Ireland, the colonization of the Irish plantations, and the policies that forced thousands of people out of Northern Ireland in the later 17th and 18th centuries could incorporated. In addition, the forced transportation of thousands of white British subjects from England, Ireland, Scotland, and Wales to the Caribbean and North American colonies might be incorporated.
Beyond this, and I think relevant, is British relocation of Black Loyalists after the American Revolution. This has been researched by many scholars, among them Simon Schama and perhaps most notably Cassandra Pybus (Epic Journeys of Freedom). The essay does mention Australia, but not in great detail. The British failure to meaningfully incorporate Black Loyalists into British society or to provide for, support, and sustain those who were relocated to Africa, Australia, and other locations certainly seems a harbinger for later attitudes. Many of these individuals went first to London, where they were seen as a prominent social problem, before their eventual relocation elsewhere. The situational parallels may be stronger in the 19th and 20th centuries with the second British empire and 19th/20th century Britain, but the 16th, 17th, and 18th century examples are valuable for tracing both English (prior to the Act of Union) and British attitudes and policy.
Without meaning to sound disparaging, parts of the essay sound a bit like contemporary political/social commentary/policy analysis (particularly where source citations are glaringly absent) while others (primarily section 3) sound like solid, if somewhat brief, historical analysis. Again, this blended format maybe approriate for a journal with an interdisciplinary social science focus, but the essay could perhaps harmonize these two elements a bit better in a more polished draft.
I do think what the author has to say in the essay is interesting, valuable, and worth publishing. I suggest a reworking that addresses the colonial aspect -- either by initially downplaying the colonial/archival focus in the abstract or by strengthening it where it is lacking in the paper. It is worth considering any relevant examples from the first British empire as well. Beyond this, I think the attention to Australia (lines 273-274) could be improved by mention of the Black Loyalists.
The writing itself is very good. There are few errors or stylistic concerns. But I recommend a little more attention to harmonizing the somewhat disparate aspects of the paper so that they flow more smoothly.
Below are some specific areas where I think additional source citations would be valuable. There is also one example where I think additional commas would improve the sentence, but this may be an individual stylistic preference.
FN1: I suggest a comma before each usage of which (deportations,' which...will, and 'deportations,' which....
Lines: 64-75: Sources?
Lines 76-89: Sources?
LIne 100: Is this the source for the preceding 10 lines?
Lines 127-134: Sources?
Line 143: What does (2002) mean? Is that a citation or date of quote?
Line 158: Should comma come after parentheses ?
Lines 189-192: Source?
Line 208: Is Gower et al the source for the entire paragraph?
Line 223: Is (2023) a source citation or date of statement or both? (See 271, 299 also)
LIne 233: Source?
Lines 237-255: Sources ?
Lines 256-263: Sources?
Line 283: Sources?
Lines 290-297: Sources?
Lines 317-320: Sources?
Lines 362-365: Sources?
Lines 382-384: Sources?
Lines 389-409: Are footnoted items the only sources?
Lines 429-433: Sources?
Lines 434-446: Sources?
Lines 429-522: Sources?
Author Response
This is an interesting and well-researched article that makes a strong contribution to scholarship.
Thank you very much for the close reading and for the extensive and constructive comments.
I do have some suggestions, however. While noting that there are six footnoted archival sources, it did seem that large portions of the paper are without source citations. And even the sections with the footnoted archival citations may be lacking some in-text source citations.
We have added more citations. Where the ‘source’ is an argument/analysis we are making, we have rephrased the sentence to make that clearer.
Given the mention in the abstract that this paper is "based on extensive archival research," I expected to see more of this up front and to also to see a little more about the research process.
Was a particular archive examined? Were particular time periods or locations examined? How were these documents found/selected?
While the documents used, and the material drawn from them, is excellent, there is an element of randomness to their presentation that makes me question how "extensive" the archival work actually was. I would like to hear more about the research process and the parameters of inquiry.
We have removed the claim of ‘extensive archival research’ (which we had removed in an earlier draft but slipped in again in error!), explained that we have relied on a synthesis of secondary literature and noted where the primary sources were found. We have also added some more historic legislation.
In addition, the abstract repeatedly stresses the colonial connection - "Colonial Echoes" is in the title -- and one reads in the abstract about colonial policies, colonial states, colonial practices.
Yet the essay does not really engage the colonial theme until page 6, and then only for about three pages. I think the colonial aspect could be more prominently presented. The word appears 16 times before section 3 (five of these in the abstract and title), a further 23 times in section 4 (beginning on page 6), about 12 times in the concluding section (beginning on page 10), and eight times in the bibliography. Given the early emphasis on it, I had expected to encounter more about the colonial context and to do so earlier in the paper. As it is, despite the earlier promise, the attention to the colonial past really receives only about 3 1/2 pages of attention in 11 pages of text, and most of that not until the second half of the essay.
It might be worth considering a slight reorganization of some material if the colonial focus/thrust is really central to the author's conception of the piece.
We have extended section three on colonial history and added more lines about it to the introduction. We experimented with interspersing it with section one but that didn’t work in terms of readability.
The essay also leans far more heavily towards the second British empire than the first. In some ways this may be appropriate, but I think there is material that could be profitably incorporated that could strengthen the argument. If the focus is solely on the later colonial period, though, this could be better explained.
We have revised the essay to include both the first and second British empires. Thank you for your suggestion.
The early British experience in Ireland, the colonization of the Irish plantations, and the policies that forced thousands of people out of Northern Ireland in the later 17th and 18th centuries could incorporated. In addition, the forced transportation of thousands of white British subjects from England, Ireland, Scotland, and Wales to the Caribbean and North American colonies might be incorporated.
Beyond this, and I think relevant, is British relocation of Black Loyalists after the American Revolution. This has been researched by many scholars, among them Simon Schama and perhaps most notably Cassandra Pybus (Epic Journeys of Freedom). The essay does mention Australia, but not in great detail. The British failure to meaningfully incorporate Black Loyalists into British society or to provide for, support, and sustain those who were relocated to Africa, Australia, and other locations certainly seems a harbinger for later attitudes. Many of these individuals went first to London, where they were seen as a prominent social problem, before their eventual relocation elsewhere. The situational parallels may be stronger in the 19th and 20th centuries with the second British empire and 19th/20th century Britain, but the 16th, 17th, and 18th century examples are valuable for tracing both English (prior to the Act of Union) and British attitudes and policy.
Without meaning to sound disparaging, parts of the essay sound a bit like contemporary political/social commentary/policy analysis (particularly where source citations are glaringly absent) while others (primarily section 3) sound like solid, if somewhat brief, historical analysis. Again, this blended format maybe approriate for a journal with an interdisciplinary social science focus, but the essay could perhaps harmonize these two elements a bit better in a more polished draft.
I do think what the author has to say in the essay is interesting, valuable, and worth publishing. I suggest a reworking that addresses the colonial aspect -- either by initially downplaying the colonial/archival focus in the abstract or by strengthening it where it is lacking in the paper. It is worth considering any relevant examples from the first British empire as well. Beyond this, I think the attention to Australia (lines 273-274) could be improved by mention of the Black Loyalists.
We have taken into account all these very useful suggestions and included mention of transportation in more detail, Cassandra Pybus’s work on the Black Loyalists, and added some more detail on our argument that colonial mobility was both ‘directed’ and also included an element of ‘immobility’ when certain groups were forcibly relocated and confined to particular places.
The writing itself is very good. There are few errors or stylistic concerns. But I recommend a little more attention to harmonizing the somewhat disparate aspects of the paper so that they flow more smoothly.
Below are some specific areas where I think additional source citations would be valuable. There is also one example where I think additional commas would improve the sentence, but this may be an individual stylistic preference.
comment |
response |
FN1: I suggest a comma before each usage of which (deportations,' which...will, and 'deportations,' which... |
added |
Lines: 64-75: Sources? |
added |
Lines 76-89: Sources? |
added |
LIne 100: Is this the source for the preceding 10 lines? |
No, the BBC reference is only for the Tony Blair announcement. The remaining data is commentary on Fig. 1 – we hope that the reference to Figure 1 makes that clear. |
Lines 127-134: Sources? |
Two sources added |
Line 143: What does (2002) mean? Is that a citation or date of quote? |
Page number added. |
Line 158: Should comma come after parentheses ? |
added |
Lines 189-192: Source? |
Two sources added |
Line 208: Is Gower et al the source for the entire paragraph? |
Yes. Gower et al reprints the MOU, which is the focus of this paragraph. |
Line 223: Is (2023) a source citation or date of statement or both? (See 271, 299 also) |
These are all dates, coming immediately after the authors. They are not direct citations but summaries of overall arguments. |
LIne 233: Source? |
This is our own analysis based on Goodwin-Gill’s argument. |
Lines 237-255: Sources ? |
This paragraph is an analysis of the court judgement referenced at the beginning of the paragraph. |
Lines 256-263: Sources? |
This is our argument, rephrased. |
Line 283: Sources? |
Have added the word ‘will’ to ‘we will show’ to indicate this is discussed later when sources are provided. |
Lines 290-297: Sources? |
Sources added |
Lines 317-320: Sources? |
Rephrased. |
Lines 362-365: Sources? |
Added another source |
Lines 382-384: Sources? |
Rephrased to show we are analysing what we have gleaned from the previously quoted source (Newman). |
Lines 389-409: Are footnoted items the only sources? |
Added a further source |
Lines 429-433: Sources? |
Rephrased to make argument clearer. |
Lines 434-446: Sources? |
This is our argument. |
Lines 429-522: Sources? |
Sources have been added here. |
Round 2
Reviewer 3 Report
The authors have made a significant revision of the previous draft. They have responded to all of my suggestions and made significant changes along the lines proposed. In my view, these enhance the essay by adding additional context and also by clarifying the various questions I had about sources and citations. The authors have done a good job satisfying my questions and concerns about the essay.
I did notice two minor issues of a typographical nature:
First, I believe source 19 by Cumper should come before source 18 by Davies in the list of works cited.
Second, there is an extra "E" (EEmpire) in line 611.
Otherwise, beyond the usual editorial process by MDPI staff for online publication, I have no further recommendations for the authors. I thank them for their careful and thorough responses to my comments.
